# Bacteria Under Metal Stress—Molecular Mechanisms of Metal Tolerance

**DOI:** 10.3390/ijms26125716

**Published:** 2025-06-14

**Authors:** Ewa Oleńska, Wanda Małek, Izabela Swiecicka, Małgorzata Wójcik, Sofie Thijs, Jaco Vangronsveld

**Affiliations:** 1Department of Microbiology and Biotechnology, Faculty of Biology, University of Białystok, 1J Ciołkowskiego Str., 15-245 Białystok, Poland; izabelas@uwb.edu.pl; 2Institute of Biological Sciences, Faculty of Biology and Biotechnology, Maria Curie-Skłodowska University, 19 Akademicka Str., 20-033 Lublin, Poland; wanda.malek@mail.umcs.pl (W.M.); malgorzata.wojcik@mail.umcs.pl (M.W.); jaco.vangronsveld@uhasselt.be (J.V.); 3Centre for Environmental Sciences, Hasselt University, Agoralaan, Buidling D, B-3590 Diepenbeek, Belgium; sofie.thijs@uhasselt.be

**Keywords:** enzymatic detoxification, exopolysaccharides, metal efflux, metallothionein, siderophores

## Abstract

Metals are natural components of the lithosphere, whose amounts and bioavailability are increasing in many areas due to their continuous release from both natural sources and intensive human activities. Some metals are essential or beneficial for living organisms, while others are non-essential and potentially toxic. When present at higher concentrations, even essential and beneficial metal ions can become harmful to all forms of life. Bacteria, unicellular organisms that have been exposed to metals since the earliest stages of life on Earth, have evolved metabolic pathways involving essential metals as well as diverse strategies to cope with metal toxicity. In the domain *Bacteria*, two main strategies have been identified: (*i*) metal exclusion, which includes cell wall sequestration and immobilization of metals in extracellular exopolysaccharides, siderophores, and other soluble microbial products, as well as (*ii*) metal tolerance, involving intracellular sequestration of metals (e.g., by metallothioneins, or low molecular weight thiols) as well as enzymatic conversion of metals to less toxic forms and/or its active efflux. Microorganisms possessing such adaptive traits are considered valuable agents for potential application in medicine, environmental sciences, and bioengineering (e.g., bioremediation and/or biomining).

## 1. Introduction

Metals, as non-biodegradable natural constituents of the Earth’s crust, are widely distributed across all compartments of the environment, including air, water, and soil [1]. Natural processes such as wind and water erosion, volcanic eruptions, geothermal activities, forest fires, and microbial activities play prominent roles in the release and redistribution of metals. The release of metals into the environment is strongly enhanced by anthropogenic activities such as metallurgy, agriculture, energy production, microelectronics, mining, sewage sludge, and waste disposal [2].

Based on their metabolic roles, metals can be classified into four main categories: (*i*) essential for life and non-toxic (e.g., calcium (Ca) and magnesium (Mg)), (*ii*) essential but harmful at higher concentrations (e.g., iron (Fe), manganese (Mn), copper (Cu), zinc (Zn), molybdenum (Mo), nickel (Ni)), (*iii*) beneficial at low concentrations (i.e., cobalt (Co), vanadium (V), selenium (Se) as well as iodine (I) and chromium (Cr)—both beneficial to plants but the latter not to microorganisms), and (*iv*) non-essential and already toxic at low concentrations (i.e., cadmium (Cd), lead (Pb), mercury (Hg)). For instance, Ca^2+^ plays a crucial role in various bacterial cellular processes such as transport through membranes, chemotaxis, cell division, as well as processes of cell differentiation (e.g., sporulation, formation of heterocysts) [3]. Mg^2+^ influences bacterial structure, cell motility, enzyme function, and cell signaling [4,5]. Fe^2+^ is a constituent of cytochromes, ferredoxin, superoxide dismutases, catalases, peroxidases, and nitrate reductases; Mn^2+^ acts as an activator for decarboxylases and dehydrogenases as well as for enzymes involved in protein, carbohydrate, and lipid metabolism; Cu^2+^ is a constituent of Cu/Zn-dismutase and nitrite reductase while Zn^2+^ plays a role in the activity and regulation of various enzymes (e.g., alcohol dehydrogenase, RNA polymerase, carboxypeptidase), proteins, DNA-DNA binding proteins, and cell metabolism [6]. Mo^2+^ is a key component of enzymes such as xanthine oxidase, sulfite oxidase, and dimethyl sulfoxide reductase [7], while Ni^2+^ is required in the prosthetic groups of [Ni/Fe]-hydrogenases, carbon monoxide dehydrogenase (CODH), acetyl-CoA synthase/decarboxylase, methyl-coenzyme M reductase, Ni-superoxide dismutase, glyoxalase I, urease, and acireductone dioxygenase [8].

Reactive and readily accessible metal ions can be harmful to all organisms, including humans. For example, metals can inhibit the activity of proteins, damage cellular membranes, compromise DNA integrity, impair DNA repair mechanisms, and cause oxidative stress. They can also inhibit enzyme activity by interfering with enzyme-substrate complex formation, altering active sites, and affecting enzyme synthesis [9,10,11,12]. As a result, metal exposure may accelerate cell death, reduce population size, and diminish the genetic polymorphism, ultimately decreasing the adaptability of affected populations [13,14,15,16,17,18].

The reactivity of metals varies and generally depends on their atomic structure and the structure of the accompanying chemical groups. To understand the stability of chemical compounds and predict the direction of chemical reactions, the theory of hard and soft acids and bases (HSAB) is often applied [19]. According to HSAB, the most stable interactions occur between hard base and hard acid atoms, which typically form ionic bonds, or between soft base and soft acid atoms, which typically form covalent bonds [20]. Ligands containing oxygen or nitrogen atoms in their structure are hard bases and bind with hard acids (e.g., IA, IIA, IIIA, IIIB, Fe^3+^). In contrast, ligands containing carbon, sulfur, or selenium are soft bases and preferentially interact with soft acids (such as Ag^+^, Cd^2+^, and Hg^2+^). Some metal ions, such as Cu^2+^ and Pb^2+^, are considered borderline acids; Pb^2+^ forms preferential interactions with soft bases like sulfur, for example, in cysteine [21,22].

Millions of years of bacterial evolution under changing environmental conditions and exposure to metal ions have led to an array of adaptive mechanisms [23]. Bacteria have evolved resistance to various metal(loid)s, including Ag^+^, AsO_2_^−^, AsO_4_^3−^, Cd^2+^, Co^2+^, CrO_4_^2−^, Cu^2+^, Hg^2+^, Ni^2+^, Pb^2+^, Sb^3+^, TeO_3_^2−^, Tl^+^, and Zn^2+^ [24]. Members of the *Bacteria* domain exhibit a wide range of adaptations, which can be categorized into two main categories: (*i*) mechanisms that prevent metals from entering the cytoplasm (exclusion strategies) and (*ii*) mechanisms that confer tolerance to metals present in the cytoplasm (tolerance strategies). Exclusion strategies involve cell wall sequestration of metals and the production of extracellular barriers that immobilize metals, such as exopolysaccharides, soluble microbial products, and siderophores. Tolerance strategies include intracellular sequestration of metals, enzymatic conversion to less toxic forms of metals, and/or their active efflux [23,25,26]. Both strategies, exclusion and tolerance, are illustrated in Figure 1.

Mechanisms of metal tolerance in members of the *Bacteria* domain have been, and continue to be, extensively investigated [2,12,26]. Since our review in 2013 [25], significant new evidence has emerged regarding the structure of operons involved in metal transport as well as the structure and function of exopolysaccharides, soluble microbial products, siderophores, and metallothioneins [12,26,27]. Although the literature provides new insights, comprehensive studies on understanding the molecular basis of metal tolerance and its regulation in microbes remain insufficient. A deeper understanding of these processes requires further investigation into the mechanisms of ion uptake and efflux, as well as the regulation of their intracellular concentrations, which ultimately determine whether metals act as nutrients or toxic agents. To present the most up-to-date data on the components of bacterial metal tolerance, we focus on a detailed analysis of bacterial metal exclusion and tolerance systems, along with the regulation of metal ion homeostasis. This knowledge is particularly relevant for its potential applications in medicine, environmental sciences, and bioengineering (e.g., bioremediation, biomining).

## 2. Cell Wall Sequestration

Cell wall sequestration refers to the complexation of metals as insoluble compounds and the accumulation of metal ions by components in the periplasm or outer membrane [27,28]. The bacterial cell wall is the first cellular structure to come into contact with metal ions present in the environment. The structure of the cell wall is a significant factor influencing the response to metals. The cell walls of Gram-positive bacteria differ from those of Gram-negative bacteria. Gram-positive bacteria are surrounded by a single thick peptidoglycan cell wall (hence termed monoderms), which is covalently linked to anionic wall teichoic acids (WTAs) and lipoteichoic acids (LTAs). Gram-negative bacteria possess a much thinner peptidoglycan cell wall, but in addition, they have an outer membrane composed of lipopolysaccharides and phospholipids (therefore called diderms).

In Gram-negative bacteria, metal binding occurs due to the presence of various functional groups on the surface, such as carboxyls, phosphomonoesters, phosphodiesters, amines, and hydroxyls [29,30]. In Gram-positive bacteria, WTAs, LTAs, and peptidoglycans play a crucial role in metal binding to the cell wall [31]. Bacterial teichoic acids are polymers, polyglycerol phosphates or polyribitol phosphates, depending on the strain, and are covalently bound to the cell wall (WTA) or anchored in the cytoplasmic membrane (LTA) [32]. The phosphodiester groups of the teichoic acids serve as the metal-binding sites. Peptidoglycan is a polysaccharide whose backbone consists of *N*-acetylmuramic acid and *N*-acetylglucosamine with peptide side chains (amino acids and diaminopimelic acid). The sites involved in metal binding by peptidoglycan are mainly the carboxyl groups [31].

For example, *Pseudomonas syringae* binds copper ions in the periplasm through the periplasmic proteins CopA and CopB and the outer membrane protein CopC, the synthesis of which is metal-dependent [27,33]. Similarly, *P. pickettii* strain US321 accumulates copper in the periplasm and the outer membrane [34]. *P. stutzeri* strain AG259 isolated from the soil of a silver mine accumulates metals as sulfide complexes on the cell surface or in the periplasm [35,36]. *Synechocystis* sp. strain PCC 6803 accumulates zinc in the periplasm [27,37]. Gabr et al. [38] reported the immobilization of lead by cell surface carbonyl, phosphate, hydroxyl, and amino groups in *P. aeruginosa* strain ASU6a. Çabuk et al. [39] reported the binding of lead by cell surface amide, sulphonamide, carboxyl, and hydroxyl groups in *Bacillus* sp. strain ATS-2.

## 3. Extracellular Barriers Immobilizing Metals

Microorganisms produce and secrete various types of compounds outside the cell that can alter metal bioavailability and, as a result, reduce metal stress. These include extracellular polymeric substances (EPSs), soluble microbial products (SMPs) [40], and smaller molecules with metal complexing properties, such as siderophores [41]. Extracellular polymeric substances are primarily exopolysaccharides (ExPS) and proteins and, in smaller amounts, lipids or nucleic acids [42]. These compounds result from the metabolism of the microorganism and its autolysis [43]. Among the EPSs, three categories of compounds are distinguished based on the strength of the binding between the cell surface and the metabolite: tightly bound EPS (TB-EPS), loosely bound EPS (LB-EPS), and soluble EPS (S-EPS) [44]. SMPs represent various compounds such as proteins, polysaccharides, humic and fulvic acids, nucleic acids, and lipids, which are secreted by the cell during metabolism, biomass growth, or degradation [45]. Both EPSs and SMPs possess functional groups—such as carboxyl, hydroxyl, or amino groups—that determine their affinity for metals. These compounds offer binding sites for metal ions and interact with them through ion exchange, complexation, or precipitation [46,47]. Due to their high affinity for metals, EPSs and SMPs play a role in stress protection and cell communication. EPSs are also involved in biofilm formation and nutrient provision [44].

### 3.1. Exopolysaccharides

Exopolysaccharides (ExPSs) are a group of extracellular polymeric substances secreted by numerous Gram-positive and Gram-negative bacteria [48]. These substances can be tightly associated with the cell wall as capsular polysaccharides (CPS) or secreted into the medium as an unattached slime layer [48]. ExPSs are composed of repeated sugar monomers, their derivatives (e.g., alcohols, acids, and aminoglycosides), and some non-carbohydrate substituents such as phosphates, acetyls, succinate, glycerol, or pyruvate [49,50,51]. The monomers in ExPS are organized in linear or branched configurations, which determine the polymer’s rheological properties. The rigid backbone of ExPS typically consists of carbohydrates connected primarily by β-(1,4)- or β-(1,3)-glycoside linkages. The α-(1,2)- and α-(1,6)-glycoside bonds between carbohydrates contribute to the flexibility of the ExPS structure [52,53]. When the repeating monomers consist of a single type of carbohydrate, such ExPSis called a homopolysaccharide. Depending on the monomers involved, it can be categorized as β-D-glucans, α-D-glucans, polygalactans, or fructans. Heteropolysaccharides are composed of different carbohydrates, e.g., D-glucose, D-galactose, L-rhamnose, N-acetylgalactosamine, or N-acetylglucosamine, or glucuronic acid [48]. ExPS are produced by microorganisms either extracellularly (outside the cell membrane and the cell wall), within the cell wall, or intracellularly [54]. Heteropolysaccharides are predominantly produced intracellularly and then transported outside the cell, while homopolysaccharides are produced extracellularly [55]. Synthesis of ExPS occurs through several mechanisms, such as (*i*) the Wzx/Wzy-dependent pathway, (*ii*) the synthase-dependent pathway, (*iii*) the ATP-binding cassette (ABC) transporter-dependent pathway, and (*iv*) the extracellular synthesis mediated by a single sucrase protein [56]. In the Wzx/Wzy pathway, monomers linked to the undecaprenol diphosphate anchor (C55) located in the inner membrane are assembled by glycosyltransferases (GTs), translocated across the cytoplasmic membrane by a Wzx protein (flippase), polymerized in the periplasmic space by the Wzy protein (polymerase), and ultimately released onto the cell surface [57]. According to the synthase-dependent pathway, the complete polymer is secreted across the cell envelope and polymerized by synthase, independently from flippase [58]. The ABC-dependent pathway is primarily involved in capsular polysaccharide (CPS) synthesis. CPSs are synthesized by GTs on the cytoplasmic side of the inner membrane as homopolymers, when a single type of the enzyme is active, or heteropolymers, when multiple GTs are involved [59]. The resulting product is exported by the complex of ABC transporter and periplasmic proteins such as PCP (polysaccharide co-polymerase) and OPX (outer membrane polysaccharide export), which are closely related to proteins of the Wzx/Wzy system. Finally, the last known mechanism, the extracellular synthesis of polymers, occurs as a result of glycosyltransferase activity that is secreted and covalently linked to the cell surface [58,60].

The diversity in ExPS structures results in a wide range of functions, including water resistance, nutrient absorption from the environment, initial attachment of cells to solid surfaces, biofilm formation, and enhanced resistance to changing environmental conditions, including metals [48]. Sequestration and removal of metals from the environment may be performed by biosorption, which utilizes living or dead microorganisms or products of their metabolism [61,62]. Perez et al. [63] demonstrated that the *Paenibacillus jamilae* strain CECT 5266 biosorbed Pb, Cd, Co, Ni, Zn, and Cu in its ExPS. Biosorption involves both physical processes, such as electrostatic interactions, Van der Walls forces, and differences in metal concentration between biosorbent and the solution, as well as chemical processes, including complexation, chelation, coordination, ion exchange, or microprecipitation [64,65]. Exopolysaccharides (ExPSs) consist of carbohydrates equipped with negatively charged functional groups (e.g., carboxyl, phosphoryl, hydroxyl) that bind to positively charged metal ions [61,66,67]. Moreover, sequestration of metal ions into bacterial ExPS may result from electrostatic interactions with negatively charged uronic acids [68], such as glucuronic, galacturonic, and mannuronic acids, as well as pyruvate [69,70]. Phosphates and sulfates further enhance the potential of EPS to immobilize toxic metal ions [71]. ExPSs effectively sequester various metals, e.g., Zn^2+^, Pb^2+^, Ni^2+^, Cu^2+^, Cd^2+^, Co^2+^, or Hg^2+^ in bacteria such as *Paenibacillus jamilae*, *Bacillus firmus*, *Bacillus licheniformis* strain KX657843, *Herbaspirillium* sp., and *Paenibacillus peoriae* strain TS7 [62,72,73,74]. Cao et al. [75] demonstrated that the ExPS-producing *Pseudomonas agarivorans* strain Hao 2018 lowered the Pb^2+^ concentration and positively influenced the *Brassica chinensis* rhizosphere microbiome, which is active in promoting plant growth. *Bacillus xiamenensis* strain PM14 and *B. gibsoni* strain PM11 produce ExPS, alleviating metal stress in *Linum usitatissimum*, while also enriching nutrients and enhancing plant development [76]. Karthik et al. [77] reported that the ExPS-producing *Cellulosimicrobium funkei* strain AR6 reduced the Cr(VI) stress of *Phaseolus vulgaris*. Mukherjee et al. [78] found that the ExPS-producing *Halomonas* sp. lowered As(III) stress. ExPS isolated from *Azotobacter chroococcum* strain XU1 adsorbed Pb^2+^ [79] as did ExPS of *Acinetobacter junii* strain L [80], *Klebsiella michiganensis* strain R19, *Providencia rettgeri* strain L2, *Raoultella planticola* strain R3, and *Serratia* sp. strain L30 [81].

ExPSs synthesized by bacteria and secreted outside the cell are key components of the extracellular matrix (ECM), which holds bacterial cells together in a biofilm community [82,83,84]. In a biofilm, a valuable niche space, the ECM is a mixture of high-molecular-weight polymers, including proteins, lipids, extracellular DNA (eDNA), exopolysaccharides, and other metabolites such as secondary metabolites [83,84]. Exopolysaccharides are the most abundant component of the ECM [85]. It is suggested that ExPSs are responsible for the ECM functions including (*i*) forming a physical-chemical barrier that protects bacteria from external factors, thereby improving their survival in various environmental conditions, (*ii*) improving the cellular adhesion to surfaces, and (*iii*) regulating the flux of nutrients and signals involved in cell differentiation [84,86,87,88,89,90].

### 3.2. Soluble Microbial Products (SMP)

Humic- and fulvic-like substances provide binding sites for metals due to their anionic functional groups, such as carboxylic, phenolic-alcoholic, and amine groups [91]. Xu et al. [92] found that humic and fulvic acids bind Cu(II) ions with a higher affinity than SMP proteins. Furthermore, these compounds also form strong bonds with Ni(II) and Se(II) [93,94]. In SMP-carbohydrates, the bonds with metals were found to be weak, but in more complex carbohydrates containing ionizable groups, such as carboxylic, phosphoric, amino, and hydroxyl groups, metals are more strongly bound [40,95]. Extracellular nucleic acids bind ions through their phosphate backbones and nucleobases [96]. Additionally, extracellular proteins of the SMP-type can interact with metals, selectively binding them to prosthetic groups, such as cytochromes, Fe-S-proteins, or metalloenzymes [97]. Alternatively, metal ion binding can occur through mutable bonds to amino acid side chains involving carboxylate groups in aspartate or glutamate residues, or sulfur in cysteine [98].

### 3.3. Siderophores

Siderophores are structurally highly diverse low molecular weight compounds (500–1500 Da) with a high selectivity and affinity for FeIII, with stability constants K_f_ > 10^30^ (e.g., Fe_(enterobactin)_^3−^ 10^49^, Fe_(bacillibactin)_^3−^ 10^47.6^) [99], which is higher than for FeII [100,101,102,103]. Brandon et al. [104] estimated the stability constant for pyridine-2,6-dithiocarboxylic acid (pdtc), a unique siderophore produced by *Pseudomonas stutzeri* and *P. putida*, which amounts to Fe_(pdtc)_^3−^ 10^33,36^, whereas Fe_(pdtc)_^2−^ 10^12^. Depending on the character of moieties involved mainly in the hexadentate coordinative binding of metals, three main categories of siderophores are distinguished: (*i*) catecholate (e.g., agrobactin isolated from *Agrobacterium tumefaciens*, parabactin from *Paracoccus* sp., fluvibactin from *Vibrio fluvialis*), (*ii*) hydroxamates (e.g., desferrioxamine E and G1 isolated from *Nocardia* sp. or *Streptomyces* sp., vicibactin from *Rhizobium leguminosarum*, or ferrirubin from *Penicillium* sp.), and (*iii*) α-hydoxycarboxylates (e.g., rhizoferrin isolated from *Zygomycetes* sp., staphyloferrin A from *Staphylococcus aureus* or vibrioferrin from *Vibrio parahaemolyticus*) [102]. Figure 2 presents some examples of siderophores. 

The transport of Fe (III) in complex with siderophores as ferrisiderophore differs between Gram-positive and Gram-negative bacteria. In Gram-negative bacteria, it is a two-stage process. First, the ferrisiderophore is detected on the cell surface and transported across the outer membrane by energy-coupled transporters, such as TonB-Dependent Transporters (TBDTs), and reaches the periplasm where the dissociation of the complex and release of iron by its reduction may occur (e.g., ferripyoverdine transport in *P. aeruginosa*) or the metal-siderophore complex is subsequently transferred straightforwardly through the inner membrane [105]. TBDTs are complexes of three transmembrane proteins that exist in various copies [106,107,108]. Iron crosses the inner membrane by means of either (*i*) permeases (e.g., ferripyochelin transport in *Pseudomonas aeruginosa*) or (*ii*) ATP-binding cassette (ABC) transporters (e.g., ferrichrome pathway in *Escherichia coli*) [109,110]. Once in the cytoplasm, iron is released through enzymatic degradation, chemical modification, or reduction from its ferric form (Fe^3^⁺) to the ferrous form (Fe^2^⁺) [111]. In Gram-positive bacteria, ferrisiderophores are detected by proteins on the cytoplasmic membrane and directly transferred into the cytoplasm through ABC transporters [112].

Apart from their high affinity for iron, siderophores can also bind other metal ions. For example, the stability constants of pyoverdine with Zn^2+^, Cu^2+^, and Mn^2+^ fluctuate between K_f_ = 10^17^–10^22^, while desferrioxamine B binds Ga^3+^, Al^3+^, and In^3+^ with K_f_ ranging between 10^20^–10^28^, while its affinity for Fe^3+^ approaches K_f_ = 10^30^ [113,114]. Braud et al. [115,116] demonstrated that pyoverdine and pyochelin released by *P. aeruginosa* form complexes with Ag^+^, Al^3+^, Cd^2+^, Co^2+^, Cr^2+^, Cu^2+^, Eu^3+^, Ga^3+^, Hg^2+^, Mn^2+^, Ni^2+^, Pb^2+^, Sn^2+^, Tb^3+^, Tl^+^, and Zn^2+^ cations. Furthermore, it has been confirmed that various metals stimulate siderophore production [109,117,118,119]. It is assumed that the chelation ability of siderophores results from the presence of lone pair electrons in functional groups. Specifically, the lone pair electrons of oxygen and nitrogen atoms in the hydroxamate group enhance the ability of metal chelation [120].

Siderophores are synthesized through three known pathways: (*i*) nonribosomal peptide synthetase (NRPS), (*ii*) nonribosomal-independent synthesis (NIS), and (*iii*) polyketide synthase (PKS) pathway [121,122]. NRPS are composed of multi-enzymatic complexes comprising: adenylation domain (A), peptidyl carrier protein (PCP), condensation domain (C), thioesterase (T), and domains involved with epimerization (E), oxidation (O), methylation (M), and cyclization (Cy) [123,124]. Synthesis begins when the A domain recognizes and activates a specific amino acid to the aminoacyl-AMP intermediate, binding it with the PCP domain where phosphopantetheine thiol is attached, resulting in an aminoacyl-S-enzyme intermediate. This intermediate is then transferred to the C domain, where it condenses with other complexes, forming a peptide bond and, as a result, extending the peptide chain [122]. The synthesis terminates in the thioesterase (T) domain, where a residue–conserved serine binds to the peptide, forming an amino ether. This is hydrolyzed, leading to the release of the mature siderophore [125]. Through the activity of nonribosomal independent synthetases, siderophores containing citric, succinic acid, or α-ketoglutarate are formed. For instance, aerobactin (*E. coli)*, achromobactin (*Pseudomonas syringae*), desferioxamine (*Streptomyces griseus*), or putrebactin (*Shewanella putrefaciens*) are products of NIS synthetase activity, which includes an acyladenylation domain that forms intermediates of dicarboxylic acids, diamines, or amino alcohols. AMP facilitates the condensation reaction with amino acid or polyamine, providing energy for the synthesis [122]. According to the modular type I PKS system, each individual module typically consists of a ketosynthase (KS) domain, an acyltransferase (AT) domain, and an acyl carrier (ACP) domain. The synthesis begins when the acyl chain—covalently attached to the ACP by the AT domain—is transferred to the active-site cysteine of the KS domain of the subsequent module. The KS domain then catalyzes the condensation reaction, extending the growing chain, which remains attached to the ACP domain. The mature product is released from the ACP domain by a thioesterase domain through reduction, hydrolysis, or cyclization. Post-synthesis modifications can be introduced by ketoreductases, dehydratases, methyltransferases, and oxidases [126,127].

Once synthesized, apo-siderophores are secreted into the environment through transporters from the major facilitator superfamily (MFS), which is a broad substrate transporter group, or efflux pumps from the RND superfamily (resistance, nodulation, and cell division). These pumps function as proton antiporters [128,129,130,131].

Siderophores produced by *Alcaligenes eutrophus* and *Pseudomonas aeruginosa* lower the toxicity of Cd, Pb, and Cu [132,133]. *P. aeruginosa* synthesizes and releases pyoverdine and pyochelin that block the absorption of Cr, Fe, Hg, and Pb [134,135].

## 4. Intracellular Sequestration of Metals

Metallothioneins (MTs) are well-studied chelators of metals involved in homeostasis of essential metals such as zinc and copper. They also capture and immobilize ballast ions in vertebrates, plants, and fungi, and, since the mid-1980s, they have also been reported in bacteria [136]. Metallothioneins are low-molecular-weight proteins capable of binding metal ions. They have been identified in a few members of the *Bacteria* domain, such as the cyanobacteria *Synechococcus* sp. PCC 7924 and *Anabaena* sp. PCC 7120, in *Pseudomonas aeruginosa*, *P. putida*, *Salmonella choleraesuis* 4A, *Proteus penneri* GM10 [137,138], and *Mycobacterium* sp. Bacterial metallothioneins (BmtAs) are referred to as SmtA when obtained from *Synechococcus* sp., PmtA from *Pseudomonas* sp., and MymT when isolated from *Mycobacterium tuberculosis* [139]. These proteins were identified through genetic analysis, and their genes are induced by metals, resulting in enhanced metal binding and sequestration. The MT genes were reported both on bacterial chromosomes and plasmids. For example, in *Anabaena variabilis* ATCC 29413, a copy of the *bmtA* gene is located on the chromosome, with another copy on a plasmid. In *Acaryochloris marina*, one copy of the gene is located on the chromosome, and two copies are found on a plasmid [140]. Metallothioneins identified in cyanobacteria are translationally synthesized peptides encoded by the *smtA* gene. Their expression is regulated by the *smtB* gene, a *trans*-acting repressor, negatively controlling the transcription of *smtA* [141]. Metallothioneins are cysteine-rich, low-molecular-weight (ca. 10 kDa) proteins that can bind metals in the cytoplasm through the thiol (-SH) groups of cysteine residues. The cysteinyl residues along the peptide chain can exist in various configurations, Cys-*x*-Cys, Cys-*x*-*y*-Cys, or Cys-Cys, where *x* and *y* represent any amino acids [142]. Because the quantity and arrangement of cysteine residues vary among bacterial MTs, these proteins can bind metals in different ways, making bacterial MTs the most variable among other known families. For example, in *Synechococcus elongatus* PCC 7942, the SmtA protein consists of 56 amino acids and binds three Zn^2+^ ions through eight cysteine and two histidine residues [143]. In contrast, PmtA consists of 70 amino acids, including ten cysteine residues and a variable number of histidines (up to three). Almost all bacteria (except *Staphylococcus epidermidis*) where MT amino acid sequences have been studied contain at least one histidine residue. In *S. elongatus* strain PCC 7942, the MT contains eight hydrophobic residues, including a pair of neighboring tyrosine residues located centrally within the protein [142]. Immobilization of lead by MT was reported in *Pseudomonas vermicola* [144], *P. aeruginosa* strain WI-1 [137], and *P. penneri* [138].

In addition to MTs, other ligands can be involved in intracellular metal ion inactivation in bacteria. They include low molecular weight (LMW) thiols such as glutathione (GSH) or bacilithiol (BSH), as well as amino acids [145,146]. Among others, GSH was implicated in immobilizing Cd^2+^ ions in *Rhizobium leguminosarum* [147,148], Cu^+^ in *Streptococcus pyogenes* [149], or Cu^+^ and Zn^2+^ excess in *Escherichia coli* [150]. Bacilithiol is the major LMW thiol in *Bacillus subtilis*, and it was found to be involved in cytosolic chelation of both Zn^2+^ [151] and Cu^+^ [152]. The role of amino acids in metal chelation is less evident; nevertheless, complexes with histidine (e.g., with Zn^2+^ in *Acinetobacter baumanii* [153]) or cysteine (e.g., with Cu^+^ in *Chlamydomonas* [154]) have been reported.

## 5. Enzymatic Conversion of Metal Ions and/or Their Efflux out of the Cell

The conversion of toxic metal into a less toxic, less available, or volatile metal-ion species represents another strategy by which bacteria handle toxic metals [24]. A well-known example of such a detoxification process is the bacterial reduction of chromium [Cr(VI)] into the less toxic Cr(III). In anaerobic bacteria, this conversion can involve two soluble, membrane-bound reductases: ChrR and YieF. The reduction of Cr(VI) into Cr(III) proceeds via a two-step process. Initially, ChrR catalyzes the reduction of Cr(VI) into Cr(V), which is subsequently reduced into Cr(III). Alternatively, YieF is capable of directly reducing Cr(VI) to Cr(III). Under anaerobic conditions, bacteria employ various membrane-bound reductases such as flavin reductases, cytochromes, and hydrogenases to facilitate such transformations [12,155].

Gavrilescu [156] demonstrated that hazardous metals can be reduced by iron- and sulfur-reducing bacteria, like *Desulfuromonas* sp. or *Geobacter* sp. Under anaerobic conditions, *G. metallireducens* reduces toxic manganese [Mn (IV)] to non-toxic Mn (II) and uranium (U) from toxic U(VI) to U(IV).

Enzymatic metal detoxification is often closely linked to metal efflux systems. The active export of toxic metal ions out of the cell, aimed at limiting their intracellular accumulation and preventing toxicity, is well-documented across different bacteria [157,158]. This variable and energy-demanding efflux system can be either energy-dependent, using ATP hydrolysis with ATPases, or chemiosmotic, based on metal diffusion. While the general mechanisms of efflux systems in microbes are quite similar, they differ in the details, depending on the specific metals or bacterial groups involved. For example, mercury or arsenic tolerance systems are homologous across the *Bacteria* domain, whereas Cd tolerance is associated with ATPases in Gram-positive bacteria and chemiosmotic cation-proton antiporters in Gram-negative bacteria [24]. Efflux pumps are genetically determined, with genes for metal homeostasis located on the bacterial chromosome, while genes involved in metal resistance are often located extrachromosomally, such as on plasmids [22,24].

The first layer of metal tolerance, based on the export of excess cations, involves members of the resistance-nodulation-cell division (RND) protein superfamily (TC 2.A.6.1.1, according to Saier’s [159] functional-phylogenetic classification for transmembrane solute transporters). In bacteria, RND proteins are involved in the transport of metals, hydrophobic and amphiphilic compounds, and nodulation factors. They also work with SecDF (protein-export membrane protein) to facilitate the transport of proteins [22]. The HME-RND (Heavy Metal Efflux—RND) family specifically handles the transmembrane transport of metals [160]. The RND protein superfamily cooperates with the membrane fusion protein family (MFP) and proteins from the outer membrane factor (OMF) family to form an efflux protein complex known as the CBA efflux system. This system mediates the transport of substrates from the cytoplasm, cell membrane, or periplasm through the outer membrane out of the cell [161]. The CzcCBA system is an example of an RND superfamily transporter. As a chemiosmotic divalent cation/proton antiporter, it deals with the efflux of Cd^2+^, Zn^2+^, and Co^2+^. The CzcCBA transporter is encoded by an operon that is transcribed tricistronically and consists of the gene *czcC*, which determines the OMF protein (an outer membrane protein), the *czcA* gene (encoding the RND protein, an inner membrane protein), and the *czcB* gene (which encodes the MFP protein that bridges the inner and outer cell membranes). The *czc* operon was first identified on the 238 kbp megaplasmid pMOL30 in *Ralstonia eutropha* strain CH34 (formerly *Alcaligenes eutropha*) [162]. In this strain, czcCBA expression is regulated by several genetic elements. Upstream to *czcCBA* are the genes *czcN* and *czcI*, which encode proteins of currently unknown function, as well as promoters *czcNp*, *czcIp*, and *czcCp*. Downstream of *czcCBA* are the *czcRS* two-component regulatory system, transcribed from the *czcDp* promoter, and the gene *czcE*. The *czcRS* consists of the inner membrane protein CzcS, which detects levels of periplasmic Zn^2+^ or Cd^2+^, and activates the response regulator CzcR. Activated CzcR functions as a transcriptional activator of *czcCBA* expression, promoting expression of the efflux pump that confers metal resistance. CzcE, a periplasmic protein, is transcribed independently of *czcRS* in response to Zn^2+^ and regulates *czcNp* expression [163].

Cobalt (CoII) and nickel (NiII) tolerance in bacteria is based on the extracellular efflux of metals and is determined by two types of operons with similar determinants: (*i*) *cnrCBA* and (*ii*) *nccCBA*. The *cnrCBA* structural region, preceded by the regulatory gene region *cnrXYH,* consists of *cnrC,* which encodes for the OMF-type protein CnrC, *cnrB,* which encodes for MFP CnrB, and *cnrA,* which encodes for the RND-type protein CnrA. The *cnr* determinant mediates bacterial resistance to Co^2+^ and Ni^2+^, while the *ncc* (nickel-cobalt-cadmium resistance) operon additionally determines tolerance to Cd^2+^. The *nccCBA* structural region, preceded by the regulatory region *nccYXH,* encodes the OMF protein NccC, the MFP-type NccB, and the RND-type protein NccA [22,164].

The second layer of metal tolerance involves the cation diffusion facilitators (CDF) superfamily (TC 2.A.4.1.1-2). One member of the CDF family, the CzcD protein, regulates the expression of the CzcCBA system in Gram-negative bacteria. As a monoprotein membrane pump, CzcD confers tolerance to Cd^2+^, Zn^2+^, and Co^2+^ even in the absence of the CzcCBA system [165]. CDF proteins primarily transport divalent cations with ions radius size 74 ± 2 pm (picometer) (equal to 0.74 ± 0.02 Å—angstrom) diameter, typically from the first transition series, e.g., Co^2+^, Cd^2+^, Fe^2+^, Ni^2+^, Zn^2+^. These proteins generally consist of about 400 amino acids and contain six transmembrane spans. Histidine is the most prevalent amino acid in CDF proteins and is predominantly located at the amino- and carboxy-termini, as well as between transmembrane helices IV and V. It is believed to play a crucial role in regulating the transport activity of the CzcD protein [98,166,167].

The third layer of the metal tolerance system, based on metal efflux, involves a superfamily of P-type ATPases, including the CPx-type ATPases family, equipped with conserved proline and cysteine amino acids [168]. As an example, the CPx-type ATPases include the CadA protein, which mediates Cd resistance in Gram-positive bacteria. The *cad* operon consists of *cadA* and *cadC* genes and was identified in *Staphyllococcus aureus* on the pI258 or pXU5 plasmids as well as on the chromosome (Tn*554*). It has also been found in *Lactococcus lactis*, *Listeria monocytogenes* (Tn*5422*), and *Bacillus firmus* [169]. Like other CPx-type ATPase members, CadA contains eight transmembrane domains with a conserved CPC (SH-C-P-C-SH) motif located in one of the domains [170]. The ATP binding domain is situated in a large cytoplasmic region that follows the transmembrane domains; the energy for metal efflux originates from ATP hydrolysis. The second *cad* operon member, the CadC protein, consists of 122 amino acids. It functions as a regulatory protein, acting *in cis* or *trans* [24].

Another example of an energy-dependent metal system is the mechanism that handles an excess of Cu [171]. Copper can replace native metal cofactors in proteins; monovalent Cu^+^ is more toxic than divalent Cu^2+^, because of the high affinity to amino acids and nucleosides [172,173,174]. The best-studied Cu-resistance systems are found in *E. coli*, where four main systems were determined: CopABCD, Cue, CusCFBA, and Pco [175,176]. The CopABCD is a P-type ATPase that contains cysteine and histidine motifs conferring high affinity for metals [177]. Four structural proteins are part of the copper resistance system: the inner membrane protein CopD, the outer membrane protein CopB, and the periplasmic proteins CopA and CopC. The Cue system serves as the first line of defense in copper tolerance and is induced at low copper concentrations [178]. It comprises the multicopper oxidase CueO and the P-type ATPase, CopA. The CopA pumps Cu^+^ from the cytoplasm into the periplasm, where CueO oxidizes cuprous (Cu^+^) into the less toxic cupric (Cu^2+^) form. Under anaerobic conditions, when CueO is inactive or when the copper concentration continues to rise, the expression of the CusCFBA system is induced in *E. coli*. This system is regulated by the two-component system CusRS. The periplasmic sensor domain of the histidine kinase CusS binds copper, activating the cytoplasm response regulator CusR. The CusCFBA system, which is encoded chromosomally, functions as a chemiosmotic efflux pump. It includes the periplasmatic metallochaperone CusF, which binds Cu^+^ and the CusCBA tripartite complex. This complex is a CBA-type transporter composed of CusA, a proton-substrate antiporter of the resistance-nodulation family (RND) located in the inner membrane; CusB, a membrane fusion protein (MFP) in the periplasm; and CusC, an outer membrane factor (OMF) [173,179]. In Gram-negative bacteria, the CusCFBA system is activated both by Cu^+^ and Ag^+^, which, under laboratory conditions, behave as soft Lewis acids. These ions exhibit high polarizability and preferentially bind to nitrogen and sulfur-containing compounds, which are soft Lewis bases [180]. Exogenous (horizontally acquired) tolerance to silver is often plasmid-encoded. For instance, the *Salmonella typhimurium* plasmid pMG101 carries the *sil* operon, which consists of seven structural genes *silABCEFP(ORF105)* regulated by the *silRS* two-component system [181,182]. Both *sil* and *pco* operons may be located on the same plasmid, forming the Copper Homeostasis and Silver Resistance Island (CHASRI).

In some *Enterobacteriaceae* species, the *pco* gene cluster has been identified on the chromosome, as in *Salmonella enterica*, or on plasmids, such as the pRJ1004 plasmid in *E. coli*. The spread and mobility of *pco* gene clusters are associated with their location on a Tn7 transposon [183]. However, the Tn7-element is associated with a *sil* gene cluster related to silver tolerance [184]. In *E. coli*, the *pco* gene cluster consists of seven genes (*pcoA*, *B*, *C*, *D*, *R*, *S*, and *E*) [171,185]. The *pcoA* codes for multicopper oxidase (PcoA) which oxidizes Cu (I) to Cu (II) in the periplasm, while *pcoC* encodes for PcoC, a periplasmic protein which binds one ion of Cu (II) [186]. *PcoE* encodes for PcoE, another periplasmic protein, which scavenges free copper ions from the periplasmic space [187]. The PcoB encoded by *pcoB* is an outer membrane protein that acts as a Cu (II) importer that may facilitate Cu uptake [188], while PcoD is an inner membrane protein of unknown function.

In case of arsenic and antimony tolerance, the *ars* operon is involved [189]. Bacteria generally take up arsenic ions through aquaglycoproteins and phosphate transporters [190]. Pentavalent arsenate As(V), the molecular analog of phosphates, is less toxic and mobile than trivalent arsenite As(III). As(V) has a high affinity and strong binding with sulphydryl groups of amino acids [191]. The activity of the *ars* operon is regulated by the ArsR protein, encoded by the *arsR* gene, which is activated by arsenates, arsenites, antimony, and bismuth. ArsR is a repressor protein, a dimer, with two cysteine residues responsible for binding arsenic. After binding with As, ArsR is released from the promoter permitting the *ars* transcription that leads to As detoxification [12]. In the *E coli* plasmid R773, another regulatory gene, *arsD,* was identified. The *arsD* encodes a *trans*-acting protein that operates independently of inductive factors [192]. Beyond the regulatory genes, the *ars* operon consists of *arsA*, *arsB*, and *arsC* genes which encode structural proteins. The *arsA* encodes a membrane-associated oxyanion-stimulated ATPase protein, *arsB* encodes a protein that functions as a chemiosmotic transporter for arsenites [As(III)] and serves as the binding site for ArsA, while *arsC* encodes arsenate reductase, an enzyme that reduces intracellular arsenates [As(V)] to [As(III)], which is then transported by ArsB. ArsA is an intramembrane ATPase which, as a dimer, binds to the membrane transporter ArsB and provides energy for As(III) efflux through hydrolysis of ATP. This efflux system is dual-functional: it can either use energy from ATP or function chemiosmotically without additional input of energy, utilizing the ArsB protein [168,193]. Several heterotrophic bacteria, such as *Alcaligenes faecalis* or *Nitrogenophaga* sp. NT-14 and the chemolithoautotrophic *Rhizobium* sp. NT-26 are equipped with arsenite oxidase, which converts As(III) into less toxic arsenate As(V) [194,195]. In *E. coli,* the *ars* operon located on R733 plasmid also provides resistance to tellurium.

Inorganic mercury (HgII) tolerance involves the *mer* operon, which is found in both Gram-positive bacteria, such as *S. aureus*, *Bacillus* sp., and Gram-negative bacteria, such as *Pseudomonas aeruginosa, E. coli*, *Serratia marcescens*, and *Acidithiobacillus ferrooxidans*. The *mer* operon enables the conversion, transport, and detoxification of inorganic mercury (by MerA) or both, inorganic and organomercury compounds (by MerB). Hg^2+^ exerts its toxicity through its high affinity to sulfhydryl groups present in cysteine, which is found in both structural proteins and enzymes. The expression of the *mer* operon is regulated by MerR [196]. In Gram-negative bacteria, *merR* is transcribed separately, in the opposite direction from the structural *mer* genes, while in Gram-positive bacteria, *merR* is positioned after the promoter/operator (P/O) site, making it the first gene in the line of *mer* genes. Following the P/O site, there are typically one to five genes, called *merT, merC, merE, merF,* and *merG*, whose products are involved in the transport of toxic Hg^2+^ from the membrane surface to the cytoplasm, where the MerA protein (a mercuric reductase, a homodimeric flavin-dependent NAD(P)-disulfide (FAD) oxidoreductase) reduces Hg^2+^ to Hg^0^ [197]. The *merA* gene locus is situated after the transport protein genes and is followed by *merB,* which encodes MerB, an organomercurial lyase enzyme that breaks the carbon-mercury bonds in organomercurial compounds, such as phenylmercury acetate (CH_3_CO_2_HgC_6_H_5_) or methylmercury (CH_3_Hg^+^). The lyase removes organic residues, releasing toxic Hg(II), which is subsequently reduced to Hg(0), a volatile mercury species, by mercuric reductase [12,198,199]. In Gram-negative bacteria, the *merB* frequency is very low; *mer* systems lacking the *merB* gene are referred to as narrow-spectrum systems, providing resistance only to inorganic mercury compounds. In contrast, systems that contain *merB* that offer resistance to both inorganic and organic mercury compounds are called broad-spectrum systems. Examples of bacterial operons involved in metal efflux illustrates Figure 3.

## 6. Regulation of Metal Uptake and Efflux in Bacteria

The dual nature of some elements, which are beneficial at low concentrations but toxic at higher levels, has driven the evolution of mechanisms that maintain intracellular metal ion homeostasis [200]. Chakraborty et al. [201] describe the existence and function of a bacterial “set-point” system, which regulates the intracellular metal concentrations. This system is chromosomally encoded and consists of: (*i*) an influx pump to scavenge metals from the environment, (*ii*) an efflux pump to remove excess ions, and (*iii*) small metal-binding proteins that shuttle metal ions from membrane transporters to cellular targets [202]. Expression of the components in this system is controlled by metal-sensing transcriptional regulators—sensor proteins that are allosteric transcription factors with metal-dependent DNA-binding activity [203]. These sensor proteins fall into seven major families of proteins, classified based on their regulatory mechanisms:

(i)three families of metal-releasable de-repressors (i.e., ArsR-SmtB, CsoR-RcnR, and CopY), which bind DNA in their *apo*-form (metal free) and release it upon metal binding;(ii)three families of metal-inducible co-repressors ((Fur–Zur and Mur), DtxR/MntR, and NikR) which bind DNA and metal simultaneously to repress transcription and dissociate from DNA in the absence of the metal;(iii)the MerR family (e.g., CueR, ZntR, PbR, CoaR), which consists of metal-activated regulators that are always bound to their DNA operator but switch from repressing to activating transcription in the presence of their cognate metal [202,204].

The metal binding affinities of these sensors define the thresholds for maintaining homeostasis. When a sensor switches from its *apo*- to *holo*-form, it can alter the expression of metal-responsive genes. The affinities for various metals generally follow the Irving-Williams series (Mg^2+^ < Mn^2+^ < Fe^2+^ < Co^2+^ < Ni^2+^ < Cu^2+^ > Zn^2+^ [203,205].

For instance, copper homeostasis in bacteria is regulated by transcriptional activators and repressors, including the MerR family members CueR and three repressors (CsoR, CopY, and ArsR) [206]. Under high Cu^+^ concentrations, CueR proteins activate the transcription of CopA and copper chaperones [207]. In contrast, at low Cu^+^ concentrations, CueR binds to DNA in a conformation that prevents interaction with RNA polymerase, thereby repressing transcription [206]. Once cytosolic Cu^+^ levels return to physiological levels, the expression of copper-resistance genes is downregulated. In different bacterial taxa, other copper sensors play a role, like CsoR in *Mycobacterium tuberculosis* [208], CopY in *Firmicutes* [209], and BxmR (an ArsR family member), in *Oscillatoria brevis* [210]. These sensors regulate distinct operons: CsoR controls the *cso* operon; CopY regulates the *copYZBA* operon, which mediates Cu^+^/^2+^ efflux; and BxmR regulates both the metallothionein (MT) synthesis as well as Cu^+^-ATPase expression.

In *E. coli,* intracellular zinc concentrations are controlled by two metal-binding regulators: Zur (a member of the Fur family) and ZntR (a member of the MerR family) [201]. Under Zn deficiency, when intracellular zinc levels are below 0.2 fM (f—femto, 10^−^^15^), Zur lacks bound metal and acts as the non-repressor, allowing transcription of Zn-uptake genes. Simultaneously, ZntR remains in its *apo*-form and represses Zn-efflux genes. When Zn levels rise above 0.2 fM, Zur becomes fully metalled, binds to promoters, and represses Zn uptake. When the Zn concentration reaches above 1.1 fM, ZntR binds Zn^2+^, changes conformation, and activates transcription of Zn genes [201,211].

## 7. Conclusions

Since their evolutionary origin, bacteria have been continuously exposed to metals. Their long-standing interaction with metals has led to the engagement of many metal ions—such as calcium, cobalt, copper, iron, magnesium, and molybdenum—into the biochemical pathways essential for metabolism and life. These prokaryotic unicellular organisms developed various mechanisms to deal with ions when they become toxic, including strategies to avoid metal toxicity by protecting cytoplasmic targets as well as mechanisms to cope with ions that traverse the cell envelope. Understanding metal tolerance mechanisms, which encompass a wide range of genetically determined adaptive traits, is of particular interest for the development of environmentally friendly technologies for remediating metal toxicity (bioremediation) as well as in the extraction of metals from ores.

Today, bioremediation and biomining applications increasingly rely on advances in bioengineering, including the use of genetically modified bacteria. These engineered strains are employed in a variety of settings, such as industrial processes, wastewater treatment, and metal recovery [135]. More recently, heavy metal biosensors have been proposed as tools to evaluate and enhance the effectiveness of bioremediation efforts [212,213]. Biosensors—based on whole microbial cells, enzymes, antibodies, or nucleic acids—are often developed through synthetic biology and are designed to selectively detect specific metals. They translate metal presence into measurable signals, such as fluorescence, optical changes, or electrical outputs [214]. To date, a variety of metal-detecting biosensors have been developed using different transcription factor families, including: MerR for mercury (Hg), CadR for cadmium (Cd), CusS and CueR for copper (Cu), ZntR for zinc (Zn), CoaR for cobalt (Co), PbR for lead (Pb), and ArsR-SmtB for arsenic (As) [12].

Beyond the benefits of understanding bacterial metal tolerance mechanisms and their potential economic applications, significant challenges remain—particularly concerning metal resistance in pathogenic bacteria. For centuries, various metals have been used in medicine as antimicrobial agents. Today, they are incorporated into a wide range of medical products, including catheters, wound creams, dressings, ointments, bandages, sprays, ear and eye drops, implants, dental treatments, vaccines, and even medically required wearable devices, often for biofilm prevention [215]. Metalloantimicrobials are also widely used in agriculture for pest and plant pathogen control, as well as in animal husbandry [215]. However, both constitutive and acquired metal resistance can develop in bacteria as a result of natural or anthropogenic selective pressures. Overcoming this resistance may require the use of multi-targeted metalloantimicrobial strategies, which exploit various mechanisms of action—such as interference with metalloenzymes, inhibition of metal-binding enzymes, disruption of membrane integrity, regulation of metal uptake/efflux systems, inhibition of bacterial persister cells, and induction of oxidative stress [216].

## Figures and Tables

**Figure 1 ijms-26-05716-f001:**
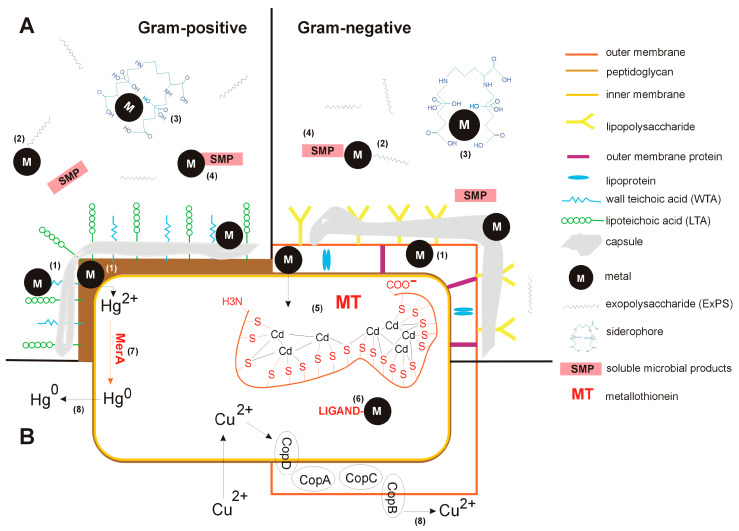
Bacterial mechanisms of metal exclusion (**A**), including cell wall sequestration (1), extracellular sequestration by exopolysaccharides (2), siderophores (3), and soluble microbial products (SMP) (4) as well as metal tolerance (**B**), including intracellular sequestration by metallothioneins (MT) (5) or other ligands (e.g., free amino acids, glutathione, or chaperones) (6), enzymatic conversion (7), and efflux of metals (8). Abbreviations: Cd—cadmium, Cu—copper, Hg—mercury, -SH—thiol group of cysteine, MerA—mercury reductase, CopA-D—proteins involved in copper transport.

**Figure 2 ijms-26-05716-f002:**
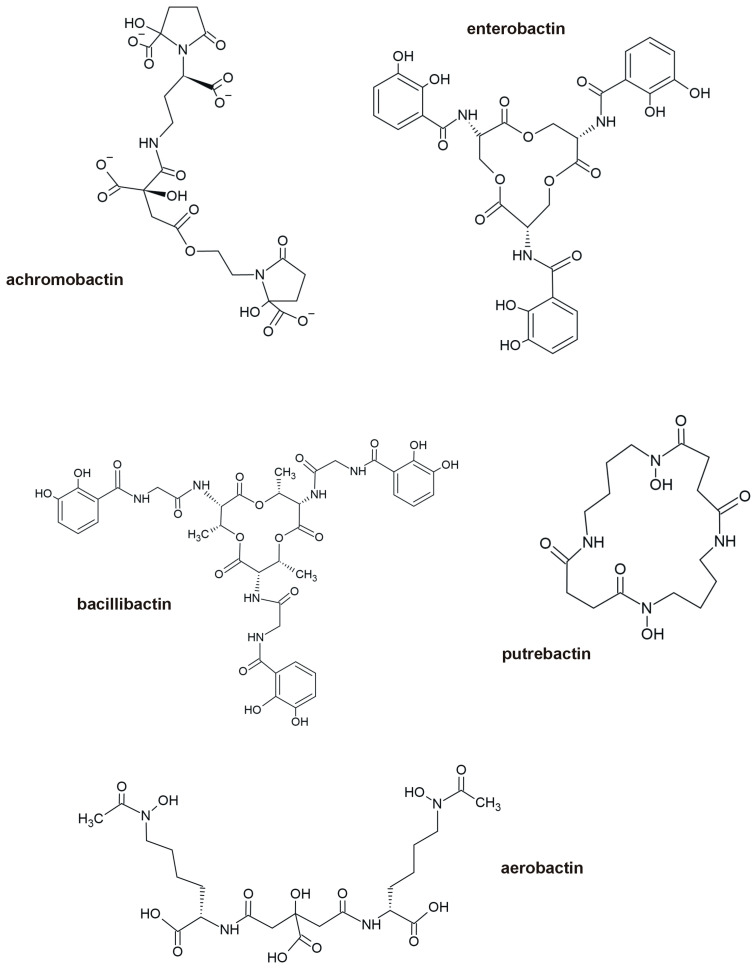
Bacterial siderophores classified by chemical structure: carboxylate type (e.g., achromobactin), catechol type (e.g., bacillibactin and enterobactin), and hydroxamate type (e.g., aerobactin and putrebactin).

**Figure 3 ijms-26-05716-f003:**
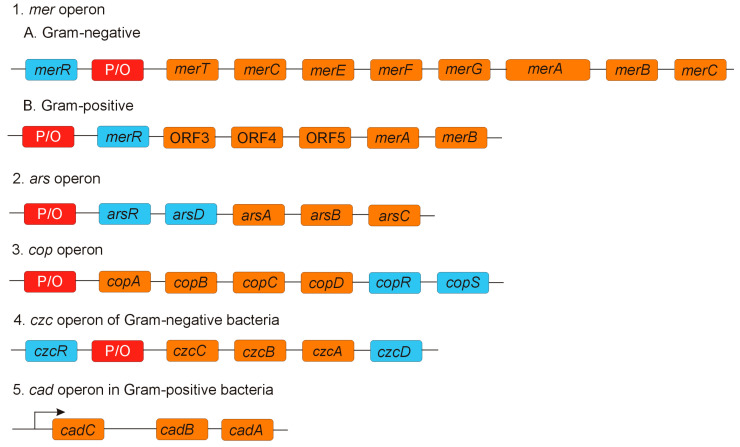
Bacterial operons involved in metal efflux. P/O—promoter/operator site. The *mer* operon: *merR* encodes the MerR regulatory protein, which activates the promoter in the presence of Hg^2+^ and represses it in its absence; *merD* encodes MerD, another regulator protein; *merA* encodes MerA, a mercuric reductase enzyme; *merB* encodes MerB, an organomercury lyase; *merC*, *mer E-G*, and *merT* encode membrane-associated proteins (MerC, MerE-G, and MerT) involved in Hg^2+^ transport from the membrane into the cytoplasm; ORF (open reading frames). The *ars* operon: *arsR* and *arsD* encode regulatory proteins ArsR and ArsD; *arsA* encodes ArsA, an intramembrane ATPase that provides energy for As^3+^ efflux; *arsB* encodes ArsB, a membrane-associated oxyanion-stimulated ATPase protein; *arsC* encodes ArsC, an arsenate reductase. The *cop* operon: *copR* and *copS* encode regulatory proteins CopR and CopS; *copA-D* encode membrane and periplasmic proteins CopA-D involved in Cu^2+^ transport. The *czc* operon: *czcD* and *czcR* encode regulatory proteins CzcD and CzcR; *czcA* encodes CzcA, an inner membrane protein; *czcC* encodes CzcC, an outer membrane protein; *czcB* encodes CzcB, a periplasmic protein that connects CzcA and CzcC. The *cad* operon: *cadA* encodes CadA, a membrane pump responsible for active Cd^2+^ efflux; *cadB* encodes CadB, which is involved in Cd^2+^ accumulation in the membrane; *cadC* encodes CadC, involved in Cd^2+^ transport into CadA; *cadD* encodes CadD, which regulates *cadA* expression. Based on [156,157,158,159,160,161,162,163,164,165,166,167,168,169,170,171,172,173,174,175,176,177,178,179,180,181,182,183,184,185,186,187,188,189,190,191,192,193,194,195,196,197,198,199,200].

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
