# Peer review of "Bacteria Under Metal Stress—Molecular Mechanisms of Metal Tolerance"

_ijms, 2025, doi:10.3390/ijms26125716_

Round 1

Reviewer 1 Report

Comments and Suggestions for Authors

Overall, the topic is interesting, however not being discussed profoundly, lacking key information. A revised framework with clearer logical hierarchy and mechanistic linkages would improve coherence and readability.

  1. In the third paragraph for the "Extracellular sequestration" section, the author mentions that some periplasmic proteins participate in metal ion interactions. However, does the periplasm strictly fall under the definition of "extracellular"?
  2. Other sections also describe periplasmic proteins involved in metal ion tolerance. This presentation approach appears inconsistent and warrants clarification.
  3. In the "efflux of the metals out of the cell" section, the author elaborates on bacterial metal effux mechanisms but fails to addree the regulatory mechanisms that activate these effux systems. Critical aspects, such as the permissible intracellular metal ion thresholds, the triggering factors for efflux pump activation under metal overload conditions, and their implications for assessing bacterial metal tolerance, are omitted. These details are essential for understanding the adaptive strategies and quantitative evaluation of bacterial resistance mechanisms.
  4. In the sixth paragraph (line 430-433) of the "efflux of the metals out of the cell" section, the descriptions "cusC is an inner membrane pump..." and "cusA is the outer membrane porin...." are factually incorrect. Author must verify such critical details throughout the paper.
  5. Additionally, the Sil system, which shares homology and functional similarity with the cus system and plays a specialized role in silver ion transport, is no discussed. Including such key information would enhance the comprehensiveness of the paper.
  6. Five sections of the paper meticulously describe diverse biochemical mechanisms underlying bacterial metal tolerance, the presented mechanisms appear isolated rather than interconnected. This limits reader to discern overarching patterns, such as how specfic bacterial groups tolerate distinct classes of metal ions. For instance, content under "Enzymatic conversion of metal ions" overlaps thematically with earlier sections, suggesting a need for sturctural reorganization.
  7. Conclusion does not adequately explore the broader implications  or future prospects of bacterial metal tolerance mechanisms. To strengthen the paper, author should emphasize potential applications in fields like medicine, bioengineering, and environmental science. 
  8. Addressing challenges, such as mitigating risks posed by metal-resistant pathogens or harnessing these mechanisms for industrial purposes, would provide a more balanced perspective and highlight the review's translational relevance.

Author Response

We would like to express our gratitude to the Reviewers for their comments concerning our manuscript entitled “Bacteria under metal stress – molecular mechanisms of metal tolerance” (Manuscript ID: ijms-3669554). All Reviewers comments are very valuable and very helpful for us in revising our manuscript. All remarks of the Reviewers have been incorporated in the revised version. The detailed responses to the comments are presented below. The changes introduced into the text were performed using the track change mode.

Comments 1 and 2: In the third paragraph for the "Extracellular sequestration" section, the author mentions that some periplasmic proteins participate in metal ion interactions. However, does the periplasm strictly fall under the definition of "extracellular"? (Comments 1) Other sections also describe periplasmic proteins involved in metal ion tolerance. This presentation approach appears inconsistent and warrants clarification (Comments 2).

Response 1 and 2: We are very grateful for this comment. We changed it to “Cell wall sequestration” (Line 110, 117-118) and were consistent through the manuscript.

Comments 3: In the "efflux of the metals out of the cell" section, the author elaborates on bacterial metal efflux mechanisms but fails to address the regulatory mechanisms that activate these efflux systems. Critical aspects, such as the permissible intracellular metal ion thresholds, the triggering factors for efflux pump activation under metal overload conditions, and their implications for assessing bacterial metal tolerance, are omitted. These details are essential for understanding the adaptive strategies and quantitative evaluation of bacterial resistance mechanisms.

Response 3: We added to the section 5 more detailed information (lines e.g., 431-442, 489-495) and presented the mechanisms of metal homeostasis as an opposite chapter 6 “Regulation of metal uptake and efflux in bacteria” (line 617).

Comments 4: In the sixth paragraph (line 430-433) of the "efflux of the metals out of the cell" section, the descriptions "cusC is an inner membrane pump..." and "cusA is the outer membrane porin...." are factually incorrect. Author must verify such critical details throughout the paper.

Response 4: We corrected this remark in lines 498-502.

Comments 5: Additionally, the Sil system, which shares homology and functional similarity with the cus system and plays a specialized role in silver ion transport, is no discussed. Including such key information would enhance the comprehensiveness of the paper.

Response 5: According the suggestion of the reviewer, we present the mentioned relationship in lines: 505-514.

Comments 6: Five sections of the paper meticulously describe diverse biochemical mechanisms underlying bacterial metal tolerance, the presented mechanisms appear isolated rather than interconnected. This limits reader to discern overarching patterns, such as how specific bacterial groups tolerate distinct classes of metal ions. For instance, content under "Enzymatic conversion of metal ions" overlaps thematically with earlier sections, suggesting a need for structural reorganization.

Response 6: Thank you for suggestion. We reorganized the structure; since the enzymatic metal detoxification is associated with efflux system, both problems were combined in one paragraph 5 (line 380) “Enzymatic conversion of metal ions and/or its efflux out of the cell”.

Comments 7: Conclusion does not adequately explore the broader implications or future prospects of bacterial metal tolerance mechanisms. To strengthen the paper, author should emphasize potential applications in fields like medicine, bioengineering, and environmental science. 

Response 7: We took into consideration the potential applications in proposed areas (lines 676-687).

Comments 8: Addressing challenges, such as mitigating risks posed by metal-resistant pathogens or harnessing these mechanisms for industrial purposes, would provide a more balanced perspective and highlight the review's translational relevance.

Response 8: We included this into main text (lines 688-702).

Reviewer 2 Report

Comments and Suggestions for Authors

A few questions and suggestions for authors:

Line 18: Check this …since the earliest stages of evolution of life on Earth

               Suggestion …since the earliest stages of life on Earth

Introduction section:

The specific aim of the review is not clearly stated at the end of the Introduction and could be summarized in a sentence. Additionally, the novel aspect of this review is not evident in this section. Including such information would help attract the reader’s interest. Also, the transition from the Introduction to Section 2 feels abrupt. A bridging sentence would help improve the logical flow.

Line 41: beneficial at low concentrations (i.e., cobalt (Co), vanadium (V), iodine (J), selenium (Se), chromium (Cr)- Not all of these elements are universally beneficial for bacteria; it would be helpful if the authors specified the context or the types of organisms for which they are considered beneficial.

Line 42: please correct iodine (J) iodine (I)

Line 55: An explanation of CO dehydrogenase is recommended at its first mention, for clarity

Figure 1: Please consider revising Figure 1 to ensure that all abbreviations and components used in the illustration (such as MT, SMP, M) are clearly explained in the legend.

Lines 120, 123, 319, 365, 366: Please consider replacing the chemical symbols Cu, Zn, Hg, As, Cd with the full terms copper, zinc… in the text

Why did the authors separate Sections 2 and 3 instead of using Section 2 as an introduction to Section 3?

The authors have previously published a review on bacterial resistance to heavy metals (Ref. [25] Oleńska & Małek, 2013). Although the current manuscript provides a broader and more detailed molecular perspective, it would be of interest for the authors to highlight how this manuscript differs from the previous work.

Author Response

We would like to express our gratitude to the Reviewers for their comments concerning our manuscript entitled “Bacteria under metal stress – molecular mechanisms of metal tolerance” (Manuscript ID: ijms-3669554). All Reviewers comments are very valuable and very helpful for us in revising our manuscript. All remarks of the Reviewers have been incorporated in the revised version. The detailed responses to the comments are presented below. The changes introduced into the text were performed using the track change mode.

Comments 1: Line 18: Check this …since the earliest stages of evolution of life on Earth. Suggestion …since the earliest stages of life on Earth

Response 1: We changed it in the abstract (line 18).

Comments 2: Introduction section: The specific aim of the review is not clearly stated at the end of the Introduction and could be summarized in a sentence. Additionally, the novel aspect of this review is not evident in this section. Including such information would help attract the reader’s interest. Also, the transition from the Introduction to Section 2 feels abrupt. A bridging sentence would help improve the logical flow.

Response 2: According the Reviewer’s suggestion we specified the aim (lines 102-104), we mentioned novel aspects (lines 94-102), and included a bridging paragraph (lines 93-106).

Comments 3: Line 41: beneficial at low concentrations (i.e., cobalt (Co), vanadium (V), iodine (J), selenium (Se), chromium (Cr)- Not all of these elements are universally beneficial for bacteria; it would be helpful if the authors specified the context or the types of organisms for which they are considered beneficial.

Response 3: We changed it and specified the types of organisms (lines 43-44).

Comments 4: Line 42: please correct iodine (J) iodine (I)”

Response 4: Has been corrected (line 43).

Comments 5: Line 55: An explanation of CO dehydrogenase is recommended at its first mention, for clarity

Response 5: We explained the abbreviation (lines 57-58).

Comments 6: Figure 1: Please consider revising Figure 1 to ensure that all abbreviations and components used in the illustration (such as MT, SMP, M) are clearly explained in the legend.

Response 6: According the Reviewer suggestion we revised it.

Comments 7: Lines 120, 123, 319, 365, 366: Please consider replacing the chemical symbols Cu, Zn, Hg, As, Cd with the full terms copper, zinc… in the text.

Response 7: We changed this according the Reviewer’s suggestion (lines 142, 144, 145, 340, 406, 407).

Comments 8: Why did the authors separate Sections 2 and 3 instead of using Section 2 as an introduction to Section 3?

Response 8: Thanks for the remark. In fact, we made a mistake in the original manuscript. We revised this part.

Comments 9: The authors have previously published a review on bacterial resistance to heavy metals (Ref. [25] OleÅ„ska & MaÅ‚ek, 2013). Although the current manuscript provides a broader and more detailed molecular perspective, it would be of interest for the authors to highlight how this manuscript differs from the previous work.

Response 9: According the Reviewer suggestion we explained this in lines: 94-102.

Round 2

Reviewer 1 Report

Comments and Suggestions for Authors

The structure and content of the draft has improved significantly.

However, in part 5, the authors mention various metal ion efflux system and functional proteins. It is recommended that the content of this section be presented in a table format.